# GENERATIVE ADAPTER: CONTEXTUALIZING LANGUAGE MODELS IN PARAMETERS WITH A SINGLE FORWARD PASS

**Tong Chen**♣*, **Hao Fang**♡, **Patrick Xia**♡, **Xiaodong Liu**♠, **Benjamin Van Durme**♡,
**Luke Zettlemoyer**♣, **Jianfeng Gao**♠, **Hao Cheng**♠†
♣ University of Washington    ♡ Microsoft    ♠ Microsoft Research

## ABSTRACT

Large language models (LMs) are typically adapted to improve performance on new contexts (*e.g.,* text prompts that define new tasks or domains) through fine-tuning or prompting. However, there is an accuracy compute tradeoff—fine-tuning incurs significant training cost and prompting increases inference overhead. We introduce Generative Adapter, an effective and efficient adaptation method that directly maps new contexts to low-rank LM adapters, thereby significantly reducing inference overhead with no need for finetuning. The adapter generator is trained via self-supervised learning, and can be used to adapt a single frozen LM for any new task simply by mapping the associated task or domain context to a new adapter. We apply Generative Adapter to two pretrained LMs (Mistral-7B-Instruct and Llama2-7B-Chat) and evaluate the adapted models in three adaption scenarios: knowledge acquisition from documents, learning from demonstrations, and personalization for users. In StreamingQA, our approach is effective in injecting knowledge into the LM's parameters, achieving a 63.5% improvement in F1 score over the model with supervised fine-tuning (from 19.5 to 31.5) for contexts as long as 32K tokens. In the MetaICL in-context learning evaluation, our method achieves an average accuracy of 44.9 across 26 tasks, outperforming the base model. On MSC, our method proves to be highly competitive in memorizing user information from conversations with a 4x reduction in computation and memory costs compared to prompting with full conversation history. Together, these results suggest that Generative Adapter should allow for general adaption to a wide range of different contexts. The code is available at
⌂ https://github.com/chentong0/generative-adapter.

## 1 INTRODUCTION

Adaptation is essential for language models (LMs) to acquire new world knowledge (Jiang et al., 2024; Hu et al., 2023; Mecklenburg et al., 2024), learn new tasks (Min et al., 2022), and personalize to individual users (Salemi et al., 2024). Existing adaptation methods typically involve either *prompting* or *fine-tuning* (Brown et al., 2020). As the scale of LMs continues to increase, adapting them becomes increasingly difficult due to efficiency constraints during both training and inference (Hu et al., 2022).

Prompting with task-specific demonstrations (*i.e.,* in-context learning (Brown et al., 2020)) or background knowledge (*i.e.,* retrieval-augmented generation (Lewis et al., 2020)) is one way to enable models to temporarily encode such relevant information, allowing flexible adaptation to various tasks. However, to maintain additional memory across sessions, some extra prompts must be added to the input, which incur an inference-time or storage overhead (Chevalier et al., 2023). Fine-tuning is another way to embed new information into the LM's parameters, retaining long-term memory. Nevertheless, it requires a training phase that is more computationally expensive than a single forward pass, and acquiring knowledge through continual pretraining has shown to be data-inefficient (Yang et al., 2024; Allen-Zhu & Li, 2024). Thus, we are interested in exploring alternative approaches for effectively and efficiently adapting pretrained LMs.

---

*Work done during internship at Microsoft Research.
†Correspondence to {chentong@cs.washington.edu, chehao@microsoft.com}

In this work, we present Generative Adapter, a novel method for training a neural network (adapter generator) to generate adapters that contextualize pretrained LMs on-the-fly with temporary knowledge from incoming contexts. Inspired by fast weights (Ba et al., 2016; Schmidhuber, 1992, *inter alia*), our approach incorporates a lightweight adapter generator on top of pretrained LM as the slow network to produce updated parameters for the fast network (the adapted LM). As far as we know, we are the first to explore this direction. Specifically, the pretrained base LM remains frozen while we train the LM-specific adapter generator to generate layer-by-layer additive updates, similar to recent parameter-efficient fine-tuning (PEFT) techniques (Houlsby et al., 2019; Hu et al., 2022). For each layer, an adapter generator network uses the outer product of past context hidden states from the corresponding base LM layer to generate delta weights. These generated delta weights are then added to the base LM weights to form an adapted LM for future predictions. Similar to previous work on fast weights, our method achieves test-time adaptation using only forward passes, allowing dynamic updates as new context arrives in sequential chunks. We train the generator end-to-end in a self-supervised manner by compressing the context into a generated adapter and then computing the next-token prediction loss on a target sequence using the adapted LM. Once trained, our method can produce adapted LMs that effectively capture knowledge from the context to solve multiple downstream tasks, thus improving the adaptability of off-the-shelf pretrained LMs.

We evaluate our method on three scenarios where on-the-fly contextualizing pretrained LMs is crucial: acquiring new factual knowledge, learning from demonstrations, and personalizing for individual users. These scenarios involve diverse forms of context with varying lengths, including documents with background knowledge, task-specific input-output examples and user-specific conversations. In the knowledge acquisition scenario, Generative Adapter effectively memorizes factual knowledge from provided documents, with minimal information loss compared to full-context prompting at short context lengths. Notably, our method excels in memorizing long-context documents, managing to handle context lengths up to 32K on StreamingQA (Liska et al., 2022) and 8K on SQuAD (Rajpurkar et al., 2016) better than continous pretraining. In learning from demonstrations on MetaICL (Min et al., 2022), Generative Adapter follows demonstrations effectively, achieving superior accuracy compared to the in-context learning of its base model. This exemplifies the model's ability to adapt to new tasks efficiently. For personalization, Generative Adapter is highly effective in retaining user information from conversations, achieving a fourfold reduction in computation and memory costs compared to full conversation prompting. In practical scenarios with many queries from the same user on edge computing devices, the benefits of our method are even more evident. This positions Generative Adapter as a highly efficient tool for personalized LMs.

Our contributions are summarized as follows:

1. We introduce Generative Adapter, a novel method for efficiently adapting pretrained LMs on-the-fly using test-time contexts. To our knowledge, we are the first to explore retaining the relevant temporary knowledge through generated parameter-efficient model updates for state-of-the-art pretrained LMs.

2. We develop an adapter generator network on top of frozen pretrained LMs to transform text contexts into updated model parameters (adapted LMs) for future queries. We also design an efficient end-to-end training process to enhance the LMs' adaptability, *i.e.,* the resulting generator augmented LM can be used for various downstream tasks using only forward passes.

3. We validate the proposed method on two representative pretrained LMs. Empirically, we show the effectiveness of Generative Adapter in various adaptation scenarios, including knowledge acquisition from documents, learning from demonstrations, and personalized user interactions. Our method proves to be generalizable across different types of contexts and applicable to multiple downstream tasks.

## 2 METHOD

We present Generative Adapter, an efficient and effective framework for directly generating additive weight updates to contextualize the pretrained LM (a *frozen* base LM) at test time. Unlike continual pretraining and supervised fine-tuning which update the pretrained LM via gradient descent, our method achieves adaptation using forward passes only. In the following sections, we first provide a task formulation and an overview of Generative Adapter (§2.1). Then we describe the the adapter

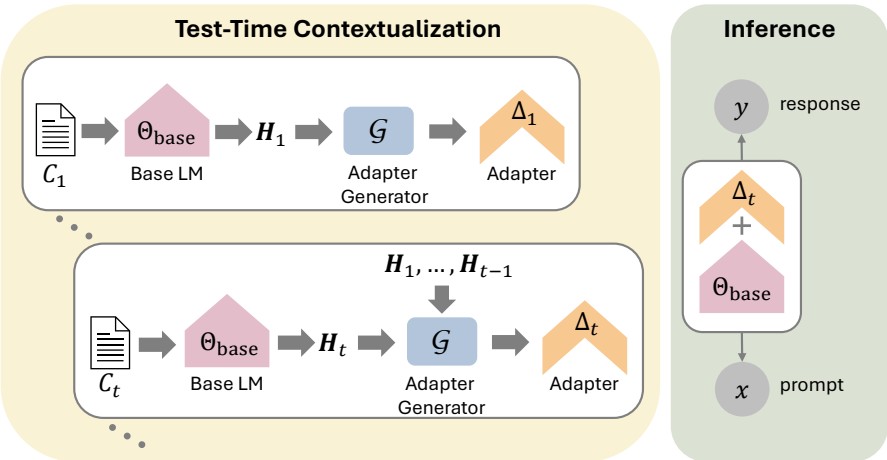

Figure 1: Overview of Generative Adapter. **Left**: During test-time contextualization, the adapters $\Delta_1, \ldots, \Delta_t$ are generated sequentially for the stream of context chunks $C_1, \ldots, C_t$. At a given time step $t$, the context chunk $C_t$ is encoded by the base LM $\Theta_{\text{base}}$ into hidden state vectors $\mathbf{H}_t$. Then the generator $\mathcal{G}$ produces a new adapter $\Delta_t$ based on the collection of hidden state vectors $\mathbf{H}_1, \ldots, \mathbf{H}_t$ representing the accumulated context. **Right**: During inference, we combine the latest adapter $\Delta_t$ with the base LM $\Theta_{\text{base}}$ to generate responses for input prompts.

generator (§2.2), followed by the self-supervised pretraining tasks (§2.3) and the normalization techniques for improving training stability (§2.4).

## 2.1 ADAPTATION WITH TEST-TIME CONTEXTUALIZATION

To *contextualize* a base model, $\Theta_{\text{base}}$, to a given context $C$, our goal is to obtain an updated model, $\Theta_C$, that can respond to user instructions using the information provided in the context $C$. In practice, the context can include different types of data, such as documents, dialogues, or task description and few-shot demonstrations.

We specifically focus on *test-time contextualization*, where context arrives incrementally as a stream of data, such as a continuous flow of documents or dialogue sessions. We represent this streaming context up to time step $t$ as $\Sigma(t) := (C_1, \ldots, C_t)$, where $C_t$ is the context chunk arriving at time step $t$. In this online adaptation scenario, the model must be efficiently adapted to each new context chunk as it becomes available.

As shown in Figure 1, we propose Generative Adapter as a framework that adapts the base model $\Theta_{\text{base}}$ to new contexts through a single forward pass as each context chunk arrives. Specifically, given test-time context $\Sigma(t)$, we adapt the base model $\Theta_{\text{base}}$ to a new model $\Theta_{\Sigma(t)}$ using a context-dependent additive adapter $\Delta_t$, *i.e.*, $\Theta_{\Sigma(t)} = \Theta_{\text{base}} + \Delta_t$. More details regarding the adapter $\Delta_t$ will be provided in §2.2. After this adaptation, the modified model $\Theta_{\Sigma(t)}$ can be utilized for any test input relevant to the context $\Sigma(t)$ during inference. For example, if the context $\Sigma(t)$ consists of a user's past conversations, the modified model $\Theta_{\Sigma(t)}$ can effectively summarize or answer questions about these conversations.

## 2.2 GENERATIVE ADAPTER

In this paper, we propose using a learned adapter generator $\mathcal{G}$ to directly produce the adapter $\Delta$ based on the streaming context $\Sigma$. The core idea is to use the adapter generator to project context token embeddings, encoded by the base language model (LM), into the matrices of each layer in the LM. Specifically, we consider only adapting the linear projection layers of the base Transformer model, *i.e.,* the key/query/value/output layers of the multi-head attention unit and the down/up projection layers of the feed-forward network.

Concretely, a linear projection layer in the $l$-th Transformer block ($l = 1, 2, \ldots, L$) can be written as $\mathbf{o} = \mathbf{W}^{(l)}\mathbf{h}$, where $\mathbf{W}^{(l)} \in \mathbb{R}^{d_{\text{out}} \times d_{\text{in}}}$ is the weight matrix, $\mathbf{h} \in \mathbb{R}^{d_{\text{in}}}$ is the input vector, $\mathbf{o} \in \mathbb{R}^{d_{\text{out}}}$

is the output vector, and we omit the bias term for simplicity. For the adapted LM, we parameterize $\mathbf{W}^{(l)} = \mathbf{W}_{\text{base}}^{(l)} + \mathbf{W}_{\Delta}^{(l)}$, where $\mathbf{W}_{\text{base}}^{(l)}$ and $\mathbf{W}_{\Delta}^{(l)}$ are the corresponding weight matrices in the base model $\Theta_{\text{base}}$ and the context-dependent adapter $\Delta$, respectively.

To generate the weights of the context-dependent adapter $\Delta$, we first encode the streaming context $\Sigma$ using the base model $\Theta_{\text{base}}$ and obtain the sequence of hidden states $\mathbf{h}_1^{(l)}, \mathbf{h}_2^{(l)}, \ldots, \mathbf{h}_M^{(l)} \in \mathbb{R}^{d_h}$ (*i.e.*, the outputs of the $l$-th Transformer block), where $M$ is the number of tokens in the context $\Sigma$, and $d_h$ is the dimension of hidden states. These hidden states are packed in a matrix $\mathbf{H}^{(l)} \in \mathbb{R}^{M \times d_h}$. Then we use the hidden states from the $(l-1)$-th Transformer block to generate the adapter's weights $\mathbf{W}_{\Delta}^{(l)}$ for the $l$-th Transformer block, *i.e.*, $\mathbf{W}_{\Delta}^{(l)} = \mathcal{G}^{(l)}(\mathbf{H}^{(l-1)})$, where $\mathcal{G}^{(l)}(\cdot) \colon \mathbb{R}^{* \times d_h} \to \mathbb{R}^{d_{\text{out}} \times d_{\text{in}}}$ denotes the layer-specific adapter generator which can transform any hidden state sequence of arbitrary length into a fixed-dimensional weight matrix. For conciseness, we will omit the superscript denoting the layer number $l$ when it does not cause ambiguity.

To obtain a generator $\mathcal{G}(\cdot)$ without the undesirable dependency on the context length $M$, we use a bi-linear function as following,

$$\mathbf{W}_{\Delta} = \mathcal{G}(\mathbf{H}) = (\mathbf{A}_1 \mathbf{A}_2)\mathbf{H}^\top \mathbf{H}(\mathbf{B}_1 \mathbf{B}_2) = (\mathbf{A}_1 \mathbf{A}_2)(\sum_{m=1}^{M} \mathbf{h}_m \otimes \mathbf{h}_m)(\mathbf{B}_1 \mathbf{B}_2), \qquad (1)$$

where $\otimes$ denotes the outer product operator, $\mathbf{A}_1 \in \mathbb{R}^{d_{\text{out}} \times d_r}$, $\mathbf{A}_2 \in \mathbb{R}^{d_r \times d_h}$, $\mathbf{B}_1 \in \mathbb{R}^{d_h \times d_r}$, $\mathbf{B}_2 \in \mathbb{R}^{d_r \times d_{\text{in}}}$ are all learnable parameters, and we set the dimension $d_r$ to be much smaller than $d_{\text{in}}$, $d_{\text{out}}$ and $d_h$ to keep the number of learnable parameters within an acceptable range.

**Dynamic Streaming Update** In practice, the context can arrive in chunks sequentially. The matrix of hidden states $H_t$ at step $t$ is computed based on all previous context chunks $\Sigma(t-1)$. This hidden state is then used to generate an adapter for the current chunk $\Sigma(t)$, which, in turn, is also used to compute the hidden states for future context steps. Based on Equation 1, to compute the adapter of $\Sigma(t)$ we need to concatenate all hidden states (*i.e.*, $[\mathbf{H}_1; \ldots; \mathbf{H}_t] \in \mathbb{R}^{(M_1 + \cdots + M_t) \times d_h}$) of the context chunks in $\Sigma(t)$ to generate the adapter, *i.e.*, $\mathbf{W}_{\Delta_t} = \mathcal{G}([\mathbf{H}_1; \ldots; \mathbf{H}_t])$.

Fortunately, our formulation allows an efficient updating mechanism without explicitly storing history hidden states, noting that

$$\mathbf{W}_{\Delta_t} = (\mathbf{A}_1 \mathbf{A}_2)([\mathbf{H}_1; \ldots; \mathbf{H}_t]^\top [\mathbf{H}_1; \ldots; \mathbf{H}_t])(\mathbf{B}_1 \mathbf{B}_2) = (\mathbf{A}_1 \mathbf{A}_2)(\sum_{i=1}^{t} \mathbf{H}_i^\top \mathbf{H}_i)(\mathbf{B}_1 \mathbf{B}_2). \quad (2)$$

Thus, the update can be efficiently computed as

$$\mathbf{S}_t \leftarrow \mathbf{S}_{t-1} + \mathbf{A}_2 \mathbf{H}_t^\top \mathbf{H}_t \mathbf{B}_1 \qquad (3)$$

$$\mathbf{W}_{\Delta_t} \leftarrow \mathbf{A}_1 \mathbf{S}_t \mathbf{B}_2 \qquad (4)$$

where $\mathbf{H}_t \in \mathbb{R}^{M_t \times d_h}$ stores the hidden states for the $t$-th context chunk, and the partial sum $\mathbf{S}_t \in \mathbb{R}^{d_r \times d_r}$ acts as the memory of history context chunks with $\mathbf{S}_0$ initialized as all zeros. Note directly storing $\mathbf{W}_{\Delta_t} \in \mathbb{R}^{d_{\text{out}} \times d_{\text{in}}}$ or $\sum_i \mathbf{H}_i^\top \mathbf{H}_i \in \mathbb{R}^{d_h \times d_h}$ would require much more memory because we control $d_r \ll \min\{d_{\text{in}}, d_{\text{out}}, d_h\}$.

Our preliminary experiments find that this architecture exhibits some empirical instability because the generated matrix $W_{\Delta_t}$ can transform an input vector $\mathbf{x}$ into a vector containing values with either extremely large or near-zero magnitudes, due to its skewed distribution of its singular values. In §2.4, we will explain how normalization can address the instability issue.

### 2.3 Learning to Update with Self-supervised Pretraining

To preserve the language modeling capability of the adapted models $\Theta_{\Sigma(t)}$ for $t \in \{1, 2, \ldots\}$, we pretrain the weight generator $\mathcal{G}$ using the next-token prediction loss of $\Theta_{\Sigma(t)}$ in a self-supervised manner on web corpora. In other words, the adapter generator is trained on top of the frozen base model $\Theta_{\text{base}}$ in an end-to-end fashion. Specifically, we use two self-supervision pretraining tasks: *reconstruction* and *completion*.

The reconstruction task (Ge et al., 2024) draws inspiration from autoencoders and aims to train the weight generator $\mathcal{G}$ to embed contextual information into the generated weights. This process

compresses the input context $(x_1, \ldots, x_m)$ into a generated adapter, $\mathcal{G}(x_{1:m})$, which is subsequently used to reconstruct the input. Formally, this is accomplished by maximizing the log-likelihood of the input tokens with the adapted LM, using weights updated from the same text: $\mathcal{L}_{\text{reconstruction}}(\mathcal{G}) = \log P(x_1, \ldots, x_m \mid \Theta_{\text{base}} + \mathcal{G}(x_{1:m}))$.

The completion task (Zhang et al., 2024; Kim et al., 2024) trains the adapted LM to generate the continuation of the given context. The goal is to maximize the log-likelihood of tokens $x_{m+1}, \ldots, x_n$, which represent the continuation of the context $x_1, \ldots, x_m$ in the dataset: $\mathcal{L}_{\text{completion}}(\mathcal{G}) = \log P(x_{m+1}, \ldots, x_n \mid \Theta_{\text{base}} + \mathcal{G}(x_{1:m}))$.

We observe using both of the task can make the generated adapter memorize and utilize the contextual information. Similar to prior work (Ge et al., 2024), the generator is trained to maximize the sum of the objective functions of the two task:

$$\max_{\mathcal{G}} \mathcal{L}_{\text{reconstruction}}(\mathcal{G}) + \mathcal{L}_{\text{completion}}(\mathcal{G}) \tag{5}$$

### 2.4 Normalization for Generated Weights

In preliminary experiments, we find that using the naive outer product for generating weights led to instability during training, causing convergence issues. When multiplying the generated matrix with the input vector, the resulting output can either diminish to near-zero or grow excessively large.

To address this instability, we introduce normalization into the formulation, *i.e.,*

$$\mathbf{W}_{\Delta_t} \leftarrow \mathbf{A}_1 \, \text{norm}(\mathbf{S}_t)\mathbf{B}_2 = \mathbf{A}_1 \, \text{norm}\left(\mathbf{A}_2 \sum_{i=1}^{t} \left(\mathbf{H}_i^\top \mathbf{H}_i\right) \mathbf{B}_1\right) \mathbf{B}_2. \tag{6}$$

Our pilot experiments find that normalization based on singular value decomposition (SVD) is particular effective, among other normalization strategies.

**SVD Normalization** The SVD normalization technique ensures the singular values of the outer product are normalized to 1. Given a matrix $\mathbf{M}$, we define SVD normalization as:

$$\text{norm}(\mathbf{M}) = \mathbf{U}\mathbf{V}^\top, \tag{7}$$

where $\mathbf{M} = \mathbf{U}\boldsymbol{\Sigma}\mathbf{V}^\top$ is the SVD factorization. This normalization resets the positive singular values of the matrix to one, preventing the vectors from excessively shrinking or exploding.

**Low-Rank SVD and LoRA** An additional benefit of SVD normalization is that it can naturally produce low-rank matrices. Instead of performing a full-rank decomposition, we approximate the input matrix with a rank-$r$ SVD decomposition, where $r$ is a hyperparameter set in advance. Consequently, the matrix can be written as the product of two low-rank matrices, similar to a LoRA adapter (Hu et al., 2022):

$$\mathbf{W}_{\Delta_t} = \mathbf{A}_1 \, \text{norm}(\mathbf{S}_t)\mathbf{B}_2 \tag{8}$$

$$= (\mathbf{A}_1 U(\mathbf{H}))(V^\top(\mathbf{H})\mathbf{B}_2), \tag{9}$$

where $U(\mathbf{H})$ and $V(\mathbf{H})$ are the matrices resulting from SVD normalization. This low-rank approximation reduces both computational cost and memory usage.

## 3 Experiments Settings

We experiment with using both Mistral-7B-Instruct (v0.2) (Jiang et al., 2023) and Llama2-7B-Chat (Touvron et al., 2023) as the base LMs. For efficiency, our main experiments train adapter generators to only update the output projection layers of the multi-head attention unit in Transformer. We study a more capable implementation in §5 and defer the full exploration of other modules for future work.

**Hyperparameters** The intermediate dimension $d_r$ and SVD rank $r$ are set to 1,024 and 128, respectively. Approximately, this leads to 500 million parameters for the generator, with the generated adapter of 32 million parameters.

**Training** Following the standard training pipeline of LM development (Jiang et al., 2023; Touvron et al., 2023), the training of our adapter generator includes a pretraining phase described in §2.3

followed by instruction tuning. For pretraining, we randomly sample 1 billion tokens from SlimPajama (Soboleva et al., 2023) which are split into segments of 8,192 tokens each. For instruction tuning, we use a mix of tasks such as question answering, in-context learning, and general instruction following, which we ensure that there is no overlap with downstream tasks, with a detailed list provided in the appendix. The context is divided into chunks of 1,024 tokens to utilize the dynamic updating mechanism described in §2.2.

# 4 MAIN RESULTS

We evaluate Generative Adapter on three representative scenarios where contextualizing pretrained LMs is crucial, *i.e.,* acquiring new factual knowledge (§4.1), learning from demonstrations (§4.2), and personalizing to individual users (§4.3).

## 4.1 DOCUMENT-BASED QUESTION ANSWERING WITH VARYING CONTEXT LENGTH

The factual knowledge stored in the parameters of a LM remains static after pretraining. Here, we consider the scenario where the model needs to adapt to new knowledge based on documents. After adaptation, it is expected to correctly answer information-seeking questions about these documents.

**Setup and Baselines** To evaluate the fact recall ability of the adapted model, we use two question answering (QA) datasets, SQuAD (Rajpurkar et al., 2016) and StreamingQA (Liska et al., 2022), where each test case consists of a passage and a corresponding question about some information from that passage. To analyze the impact of context length on performance, we conduct an evaluation using contexts of varying lengths.

We divide the documents in the corresponding test set evenly into groups, with each group having an average length of $k$ tokens ($k \in \{512, 1K, 2K, 4K, 8K, 16K, 32K\}$). Thus, the model should contextualize on the article in each group and evaluate fact recall by the question associated with the articles. The QA accuracy is evaluated by comparing the generated output with the gold answer for all questions associated with the documents within the group. Following Rajpurkar et al. (2016), F1 score is used as the metric for evaluation.

We also analyze the computational and storage requirements of Generative Adapter, which comprises three phases: general-purpose pretraining, contextualization, and inference. The generator is pretrained once and can subsequently be used for any task. During the contextualization phase, Generative Adapter encodes the context into an adapter with a single forward pass. In the inference phase, the adapted model generates responses based on the input. Beyond the LM parameters, the extra storage required includes the parameters of the generative adapter.

Here, we consider both full parameter fine-tuning and full context prompting using the same base model as baselines. For fine-tuning, we consider two variants. The first approach, *supervised fine-tuning* (SFT), trains the base model exclusively on a training set of question-answer pairs sourced from articles distinct from those in the test set. The second variant, known as *continual pretraining* (CPT), involves first training the base model on all documents in the test set, followed by further adaptation through SFT using the the training set of question-answer pairs. During inference, we evaluate the fine-tuned model in a closed-book manner, *i.e.,* the model is tasked with directly producing the answer to a given question. For prompting, we simply concatenate all documents in the group as a single context and prompt the base model to respond accordingly. Specifically, for Llama2-7B-Chat, if the context length exceeds the maximum limit of 4K tokens, we truncate the prompt to include only the last 4K tokens. For Generative Adapter, we create an adapted model for each document group, which is similar to how the context is encoded as prompting. After that, the adapted model is asked to answer the question again in a closed-book fashion, akin to fine-tuning.

**Results** We present the QA accuracy results for SQuAD and StreamingQA and the computation costs for StreamingQA in Figure 2 and Figure 3, respectively. Both fine-tuning methods (SFT and CPT) are evaluated in a closed-book manner, resulting in constant QA performance regardless of varying context lengths. In contrast, both Generative Adapter and prompting are evaluated on varying context lengths, where recalling facts can become more difficult as the context length increases.

As expected, both our method and prompting achieve improved QA performance by using relevant contexts compared to supervised fine-tuning baselines. Notably, Generative Adapter is highly effec-

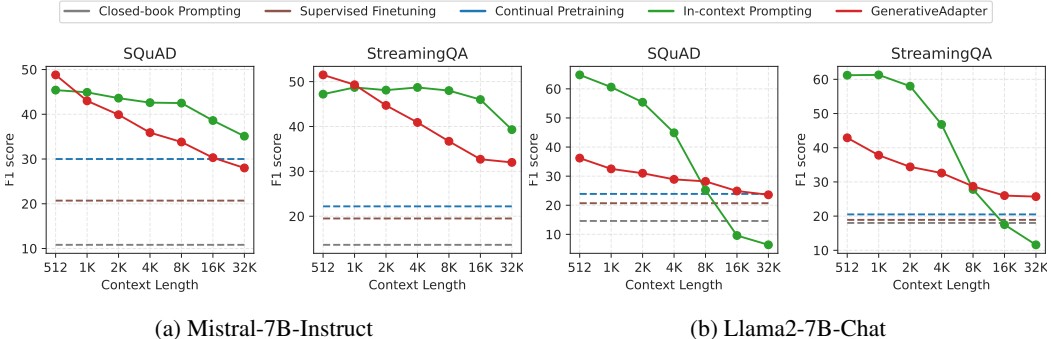

Figure 2: Document-based QA performance of Generative Adapter across different context lengths. Both fine-tuning methods (supervised fine-tuning and continual pretraining) are evaluated in a closed-book manner and remain consistent F1 across context lengths. Generative Adapter achieves the same inference time as fine-tuning methods while demonstrating higher knowledge recall.

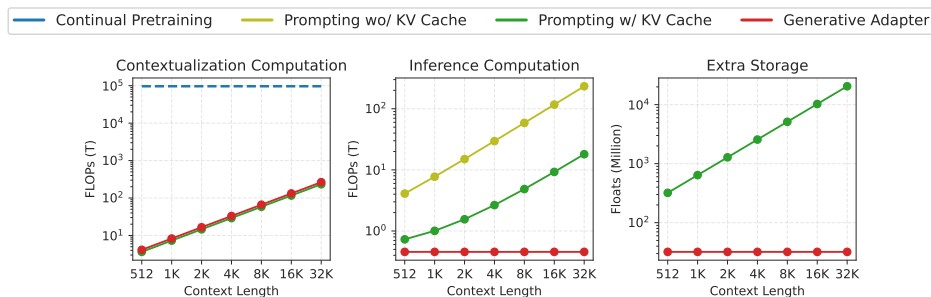

Figure 3: Computation and storage requirements for Generative Adapter and baseline methods on StreamingQA. For Generative Adapter, the context is converted into an adaptor during contextualization and then stored for inference. For the prompting method, the key-value (KV) cache can be generated during contextualization and reused during inference.

tive when the context is relatively short ($< 1K$ tokens). Moreover, it avoids the additional inference overhead associated with prompting, which requires attention computation over the context input regardless of using key-value (KV) caches (illustrated by the green lines in Figure 3). This overhead issue worsens with longer contexts.

In most cases, Generative Adapter outperforms CPT, especially when the context length is less than 8K tokens. Importantly, although both approaches adapt model parameters using documents, Generative Adapter requires preprocessing time (forward passes only) that is orders of magnitude smaller than CPT (which involves multiple forward and backward passes), as demonstrated in Figure 3.

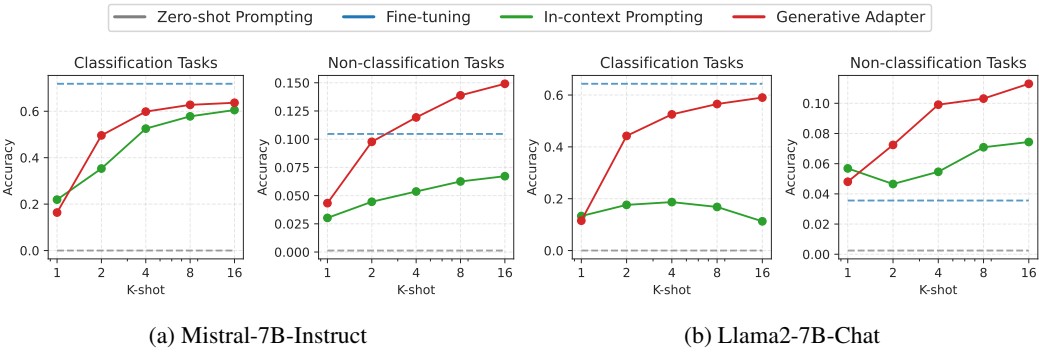

Figure 4: Accuracy plots on MetaICL with varying K-shot in-context examples. Both fine-tuned and zero-shot prompting baselines are instructed to complete the task without any in-context examples.

## 4.2 IN-CONTEXT LEARNING WITH VARYING IN-CONTEXT EXAMPLES

In the prompting paradigm, one emerging ability of pretrained LMs is that they can perform a task with a few task-specific input-output examples as context on unseen cases, also known as in-context learning (Brown et al., 2020). Here, we are interested to see whether Generative Adapter can provide further benefits in enhancing the base LM's in-context learning ability.

**Setup and Baselines** We conduct experiments using MetaICL (Min et al., 2022), consisting of 26 test tasks. We also ensure that none of these test tasks were seen during the training of adapter generator. For each task, we use 1, 2, 4, 8, and 16 demonstrations randomly sampled from the corresponding development split following the MetaICL evaluate pipeline. To reduce evaluation variance, we repeat the sampling process five times for each few-shot setting. We report separate average accuracy for classification and non-classification tasks. For classification tasks, achieving high accuracy requires the model to learn both the candidate options and the input-output relationships from the provided examples. For non-classification tasks, the model also needs to learn the output style.

We consider three baselines: 1) zero-shot prompting with the base LM where only task instruction is provided; 2) standard few-shot prompting with the base LM (Min et al., 2022) where each test case is prepend with those few-shot examples; and 3) fine-tuning the base LM on each evaluation task using 16 input-output pairs, corresponding to the maximum number of shots in our evaluation.

**Results** Figure 4 summarizes the results for MetaICL with various number of in-context examples for both classification and non-classification tasks. The performance of fine-tuned models (blue lines) and zero-shot prompt baselines (grey lines) is evaluated without demonstrations, resulting in constant performance across different numbers of shots. While fine-tuned models generally achieve higher accuracy on classification tasks, their performance on non-classification tasks is lower. We speculate that the few-shot setting (16 shots) is insufficient for the model to learn the desired output style through fine-tuning. In contrast, Generative Adapter outperforms few-shot prompting in most cases, with more significant improvements observed in the more challenging non-classification tasks, where the model must adapt to specific output styles. This indicates that the generated adapter not only retains the in-context learning ability but also enhances the base model.

## 4.3 PERSONALIZATION

Using LMs to analyze users' behaviours and memorize their preferences is the key to unlocking a tailored and engaging user experience, *i.e.,* personalized LMs. Towards this goal, we focus on evaluating the LM's ability to memorize user information in conversations.

**Setup and Baselines** We use the Multi-Session Conversation (MSC) dataset (Xu et al., 2022) for our experiments, following Packer et al. (2024). Each test case comprises a multi-session human-human conversation between two participants, along with a question regarding information mentioned within the conversation. The average length of the conversational context is 2.5K tokens, which makes it inefficient to prompt the model repeatedly with the entire conversation history for the same user. Similar to document-based QA (§4.1), we evaluate the model quality using the F1 score by comparing the generated answers to the ground truth. We also report computation and memory costs. Here, we use Mistral-7B-Instruct as the base LM.

As baselines, we include both closed-book and full-conversation prompting based on the base LM, where the former involves random guesses and the latter incurs higher computation and memory costs by storing the entire long conversation. We also include the state-of-the-art prompt compression method, UltraGist (Zhang et al., 2024), which reduces the context into fewer token embeddings, thereby saving computation and memory costs.

**Results** The results on MSC are summarized in Table 1. As expected, the closed-book approach, which does not memorize any user information performs very poorly. In contrast, methods that utilize proper user conversations as context can accurately recall user information, achieving reasonable answer accuracy. Although using the entire conversation leads to better accuracy, full conversation prompting incurs significant computation and storage costs, *i.e.,* 4x those of Generative Adapter. Such costs are highly undesirable for personalizing LMs for individual users, especially since most computations occur on edge devices without power GPUs. Comparing to UltraGist at the same level of storage cost (compressed into 512 tokens), Generative Adapter further reduces inference

Table 1: Performance comparison on MSC. A higher F1 indicates better performance, and lower inference computation and extra storage costs are preferable. For Ultragist (Zhang et al., 2024), fewer compressed tokens (noted in parentheses) correspond to lower computation and memory costs.

| Model | F1 | Inference Computation (TFLOPS) | Extra Storage (M floats) |
|---|---|---|---|
| Closed-book | 8.1 | 0.505 | 0 |
| Full-conversation Prompting | 66.0 | 2.059 | 128+ |
| Ultragist (64 Tokens) | 26.5 | 0.514 | 4 |
| Ultragist (128 Tokens) | 32.2 | 0.552 | 8 |
| Ultragist (256 Tokens) | 38.3 | 0.627 | 16 |
| Ultragist (512 Tokens) | 40.8 | 0.772 | 32 |
| Ultragist (1K Tokens) | 44.4 | 1.067 | 64 |
| Ultragist (2K Tokens) | 42.4 | 1.658 | 128 |
| Generative Adapter | 40.2 | 0.505 | 32 |

cost without performance drop. In real world scenarios with many queries from the same user, the benefits of our method are even more pronounced.

# 5 ANALYSIS

## 5.1 MODEL DESIGN OPTIONS

Here, we exam different Generative Adapter design choices. Specifically, we train adapter generators for Mistral-7B-Instruct under various configurations and assess their quality based on reconstruction and completion perplexities on the validation set. Table 2 summarizes the results. These metrics have shown a strong correlated with model quality, *e.g.,* the default setting (row 1) achieves an F1 score of 40.2 on MSC, while using the Frobenius norm (row 4) reduces the score to 27.1.

**Mixing pretraining tasks improves generalization.** Training only on one task degrades performance. Without the completion task, completion perplexity deteriorates, suggesting overfitting to memorization. This highlights its role as a regularizer, helping Generative Adapter distill contextual information into adapters for better future predictions.

**SVD is a more effective normalization.** We compare SVD-based normalization (default) to Frobenius norm. While computationally simpler, Frobenius norm exhibits inferior performance, likely due to excessive shrinkage in certain directions, reducing model expressiveness.

**More updatable parameters improve performance.** By default, we insert adapters in the attention output projection layer. Switching to the feedforward down-projection layer (tripling the number of updated parameters) enhances both perplexities. Due to computational constraints, we leave further exploration to future work.

## 5.2 COMPATIBILITY WITH BASE LM AND RAG

First, we exam whether Generative Adapter preserves the capabilities of the base model by adding the adapter layers. We generate an adapter using the prompt "You are a helpful AI assistant" and evaluate it on MMLU (0-shot) (Hendrycks et al., 2021). The base model (Mistral-7B-Instruct-v0.2) scores $0.574$, and Generative Adapter achieves $0.576$, indicating a negligible impact on accuracy.

Generative Adapter can also be seamlessly combined with RAG (Lewis et al., 2020). To illusrate this, we combine Generative Adapter with RAG by prepending the most relevant 100-token chunk (retrieved via BM25) to the query at inference. On StreamingQA with a 1K-token context, Generative Adapter alone achieves an accuracy of 49.3, while Generative Adapter + RAG reaches 63.6 with only 0.1K additional tokens. In comparison, full-context RAG requires 1K tokens to achieve 67.8 accuracy. This highlights the effectivenss of using Generative Adapter alongside RAG to enhance performance with minimal additional context. The full results are shown in Table 6 of Appendix.

Table 2: The validation set perplexity of the pretrained model under different design choices.

| Factor | Setting | Reconstruction Perplexity | Completion Perplexity |
|---|---|---|---|
| - | Default | 1.75 | 7.40 |
| Pretraining Task | Reconstruction Only | 1.75 | 34.34 |
| | Completion Only | 6.38 | 6.71 |
| Normalization | Frobenius | 7.72 | 7.32 |
| Module | Feedforward | 1.68 | 7.26 |

# 6 RELATED WORKS

**Fast Weights**: Our proposed method is closely related to the idea of "fast weights" (Hinton & Plaut, 1987; Ba et al., 2016; Schlag et al., 2021), which makes the model weights being adaptive to the model input. Context-dependent fast weight programmers (FWPs) introduced by Schmidhuber (1992; 1993) use a slow network with slow weights to reprogram the fast weights of the corresponding fast network. Schlag et al. (2021) point out that self-attention without softmax and other linear Transformer variants (Tsai et al., 2019; Katharopoulos et al., 2020; Choromanski et al., 2021; Peng et al., 2021) can be viewed as FWPs. Clark et al. (2022) propose fast weight layers which are added on top of the Transformer model after the last attention layer for language modeling. Different from previous work mainly focusing on specific tasks, our goal is to enhance frozen pretrained LMs with fast associative memory for general language processing. Instead of using a slow network to program a separate fast model, our method can be viewed as a self-programming model, *i.e.,* context encoded by the base LM is used to update the base LM itself. Our work is also related to hypernetworks (Ivison et al., 2023; Vladymyrov et al., 2024; von Oswald et al., 2020), which typically introduce additional layers to improve multi-task learning and in-context learning. However, our method directly integrates generative fast weights into the Transformer architecture, and our method can recall user-provided facts for tasks such as question answering beyond in-context learning.

**Adapting LMs via Meta-Learning**: Recent work explores adapting pre-trained LMs to an online stream of documents using meta-learning. Hu et al. (2023) propose context-aware meta-learned loss scaling, which reweights token-level losses during online fine-tuning, addressing the inefficacy of naive fine-tuning for downstream QA. Tack et al. (2024) introduce a meta-learned amortization network that predicts parameter-efficient fine-tuning modulations for individual context documents, which are then aggregated for QA. Unlike these approaches, which typically require a nested training loop, our adapter generator augments pre-trained LMs and enables end-to-end training with self-supervised objectives.

**Parameter-Efficient Fine-Tuning (PEFT)**: Generative Adapter employs a low-rank adapter akin to LoRA (Hu et al., 2022), which was originally designed for PEFT. Several derivatives of LoRA exist such as AdaLoRA (Zhang et al., 2023) and DoRA (Liu et al., 2024), along with various other PEFT strategies such as serial adapters (Houlsby et al., 2019) and prefix tuning (Li & Liang, 2021). A thorough survey of PEFT methods is presented by Han et al. (2024). Most work focuses on task-specific fine-tuning scenarios. Instead, Generative Adapter is a general LM and does not require a downstream dataset for adaptation.

# 7 CONCLUSION

In this work, we introduce Generative Adapter, a method for efficiently adapting pretrained LMs on-the-fly using test-time context through forward passes only. We design an adapter generator network on frozen pretrained LMs to transform text contexts into updated model parameters. Trained end-to-end with the frozen LM using two self-supervised tasks on web corpora, Generative Adapter is evaluated in three scenarios: acquiring new factual knowledge, learning from demonstrations, and personalizing to individual users. Our experiments show that Generative Adapter reduces information loss compared to continual pertaining in retaining factual knowledge from new documents. Additionally, the model effectively adapts to new task instructions when learning from demonstrations. Finally, Generative Adapter achieves comparable fact recall to efficient prompting methods while using less inference-time computation, making it well-suited for personalization. Future work could explore scaling the adapter generator, such as integrating adapters into additional layers, and investigating more selective update rules (Schlag et al., 2021).

ACKNOWLEDGMENTS

We thank the members of the Deep Learning Group at Microsoft Research and Semantic Machines for their invaluable discussions and support. We also appreciate the insightful feedback provided by members of Hannaneh Hajishirzi's and Luke Zettlemoyer's labs.

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

# A  DATASET DETAILS

**Pretraining.**  We pretrain our models using a randomly sampled subset of 1B tokens from the SlimPajama corpus. For validation, we sample an additional 100 segments, each containing 2K tokens, from the same corpus.

**Instruction Tuning.**  We perform instruction tuning using a combination of question answering, in-context learning, and instruction following datasets, following prior studies (Lin et al., 2024; Ge et al., 2024; Zhang et al., 2024).

# B  TRAINING SETUP

**Implementation**  We empirically found that normalization is crucial for Generative Adapter to function effectively. For SVD normalization, we implemented it using `torch.svd_lowrank()`, setting the number of iterations to 1.

Generative Adapter is able to generate the adaptors for prefixes of chunks simultaneously by processing the context chunks in parallel. The computation proceeds by processing the hidden states of each Transformer block for all context chunks layer by layer. Given the hidden states of $\Sigma(1), \ldots, \Sigma(t)$ from the $(l-1)$-th Transformer block, denoted by $H_{1:t}^{(l-1)}$, we first compute the accumulated outer product $S_{1:t}^{(l)}$ using Equation 2. We then normalize this outer product to obtain the additive matrix $W_{\Delta,1:t}^{(l)}$ using Equation 6, and finally get the output of the $l$-th Transformer block $H_{1:t}^{(l)}$ by the base model.

**Pretraining.**  For Mistral-7B-Instruct (hereafter referred to as Mistral), we use a learning rate of $5 \times 10^{-5}$, and for Llama2-7B-Chat (Llama2), we use $1 \times 10^{-4}$. We apply a weight decay of 0.01 and no dropout. The adapter added to the base model are scaled by $1/16$ for Mistral and $1/8$ for Llama2. We employ a WarmupDecayLR learning rate scheduler with a 100-step warmup and use the Adam optimizer. The global batch size is set to 8. Pretraining the adapter generator on 1B tokens takes approximately 20 hours using 8 NVIDIA H100 GPUs.

**Instruction Tuning.**  For instruction tuning, we largely follow the same configurations as in pretraining, with some adjustments. We set the learning rate to $5 \times 10^{-5}$ for both Mistral and Llama2 models. We train the models for 2 epochs and use a batch size of 32.

# C  EXPERIMENT SETUP

**Document-based QA.**  We set up experiments for document-based question answering (QA) using both supervised fine-tuning and continuous pretraining. For supervised fine-tuning on question-answer pairs, we train on the training split of each dataset, evaluate on a validation set, and employ early stopping when the validation loss increases. We use a learning rate of $1 \times 10^{-5}$ and a global batch size of 64. For continuous pretraining, we train for 8 epochs using the log-likelihood of the document as the training loss, with learning rates of $1 \times 10^{-5}$ for Mistral and $3 \times 10^{-5}$ for Llama2. Each passage is treated as a training sample, and we use a global batch size of 16.

For closed-book prompting and in-context prompting, we apply an instruction template to encourage the model to generate a short answer. The prompts are shown in Figure 7.

**In-Context Learning.**  We explore in-context learning using both fine-tuning and prompting methods. For fine-tuning, we conduct task-specific fine-tuning on 16 samples for each dataset. We use a learning rate of $5 \times 10^{-6}$ for Mistral and $1 \times 10^{-5}$ for Llama2. A validation set of 16 samples, disjoint from the training set, is collected from the same dataset. We train the model for a maximum of 40 epochs, employing early stopping if the validation loss increases for three consecutive epochs.

For in-context prompting, we observe that omitting additional instructions yields better performance for Mistral, whereas adding an instruction template improves performance for Llama2. The prompts are shown Figure 7.

Table 3: Statistics of data used in the instruction tuning.

| Type | Dataset | #Docs | #Instructions | Context len | Instruction len | Response len |
|------|---------|-------|---------------|-------------|-----------------|--------------|
| Question Answering | COQA (Reddy et al., 2019) | 1798 | 57.2 | 1083.5 | 5.5 | 2.7 |
| | DROP (Dua et al., 2019) | 1379 | 38.4 | 848.6 | 11.0 | 1.4 |
| | NarrativeQA (Kočiský et al., 2018) | 1047 | 29.4 | 574.5 | 8.5 | 4.4 |
| | PubMedQA (Jin et al., 2019) | 1000 | 1.0 | 200.2 | 12.9 | 40.7 |
| | Quail (Rogers et al., 2020) | 560 | 16.2 | 332.7 | 8.7 | 4.9 |
| | MS MARCO (Bajaj et al., 2018) | 4832 | 16.5 | 1152.1 | 6.0 | 14.0 |
| In-context Learning | MetaICL (Min et al., 2022) | 11888 | 3.5 | 1776.8 | 84.3 | 2.9 |
| Instruction Following | BookSum (Kryscinski et al., 2022) | 2914 | 1.0 | 1158.6 | 7.0 | 205.3 |
| | PwC (Ge et al., 2024) | 13102 | 12.4 | 348.1 | 10.3 | 23.3 |

## D  ADDITIONAL RESULTS

This section presents extra results for document-based question answering and in-context learning evaluation.

### D.1  DOCUMENT-BASED QA

In Section 4.1, we evaluate the knowledge recall capability of Generative Adapter and baseline models. The complete numerical results are provided in Table 5.

We also assess the performance of Ultragist (Zhang et al., 2024) on document QA. Since Ultragist requires a predefined compression rate for token reduction, a direct comparison with our method is not easy. To address this, we report results for compression ratios of 2, 8, and 32, as shown in Table 7 and Figure 8. As expected, Ultragist's performance degrades significantly as the compression ratio increases.

Unlike Ultragist, Generative Adapter modifies the model parameters directly, ensuring that its inference time remains identical to that of the base model after contextualization. In contrast, Ultragist's inference and storage costs depend on the number of gist tokens, which is approximately equal to the original context length divided by the compression ratio. Given its efficient inference, Generative Adapter is particularly suitable for scenarios where the model undergoes contextualization once and is then reused multiple times, making it an effective solution for resource-constrained environments such as edge computing.

### D.2  IN-CONTEXT LEARNING

Expanding on Section 4.2, we further evaluate Ultragist on MetaICL, with results summarized in Table 8. Ultragist performs worse than in-context prompting on classification tasks but shows a slight improvement on non-classification tasks. In contrast, Generative Adapter consistently outperforms both Ultragist and in-context prompting across various tasks, demonstrating its effectiveness.

### D.3  RETRIEVAL-AUGMENTED GENERATION

Generative Adapter can be combined with prompting techniques or other prompt compression methods to provide complementary benefits. To illustrate this, we integrate Generative Adapter with Retrieval-Augmented Generation (RAG). In this hybrid approach, the contextualization phase remains unchanged, and at inference, we prepend the most relevant 100-token chunk (retrieved using BM25) to the query. Additionally, for comparison, we report results where the entire context is prepended to the query (denoted as "Generative Adapter + Context"). The results are presented in Table 6.

Table 4: Training and test datasets of MetaICL.

| | |
|---|---|
| Train | piqa, hate_speech_offensive, google_wellformed_query, social_i_qa, circa, quoref, glue-sst2, scitail, emo, cosmos_qa, freebase_qa, ag_news, art, paws, kilt_ay2, glue-qnli, quail, ade_corpus_v2-classification, sciq, hatexplain, emotion, glue-qqp, kilt_fever, kilt_nq, dbpedia_14, kilt_zsre, hellaswag, squadwith_context, hotpot_qa, glue-mnli, ropes, squad-no_context, kilt_hotpotqa, discovery, superglue-record, race-middle, race-high, lama-trex, swag, gigaword, amazon_polarity, biomrc, tab_fact, tweet_eval-emoji, tweet_eval-offensive, tweet_eval-sentiment, tweet_qa, imdb, lama-conceptnet, liar, anli, wiki_qa, kilt_trex, wikisql, wino_grande, wiqa, search_qa, xsum, yahoo_answers_topics, yelp_polarity, yelp_review_full |
| Test | quarel, financial_phrasebank, openbookqa, codah, qasc, glue-mrpc, dream, sick, commonsense_qa, medical_questions_pairs, quartz-with_knowledge, poem_sentiment, quartz-no_knowledge, glue-wnli, climate_fever, ethos-national_origin, ethos-race, ethos-religion, ai2_arc, hate_speech18, glue-rte, supergluecb, superglue-copa, tweet_eval-hate, tweet_eval-stance_atheism, tweet_eval-stance_feminist |

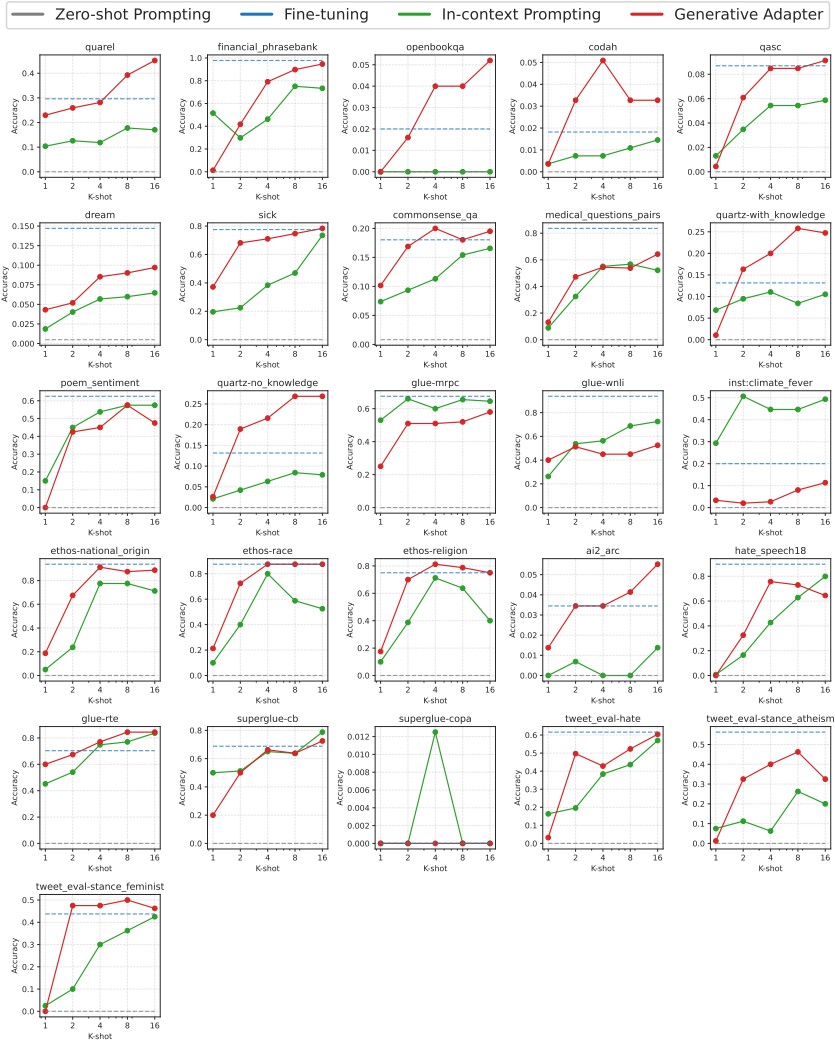

Figure 5: In-context learning evaluation of Generative Adapter, based on Llama2-7B-Chat, across 26 test datasets from MetaICL.

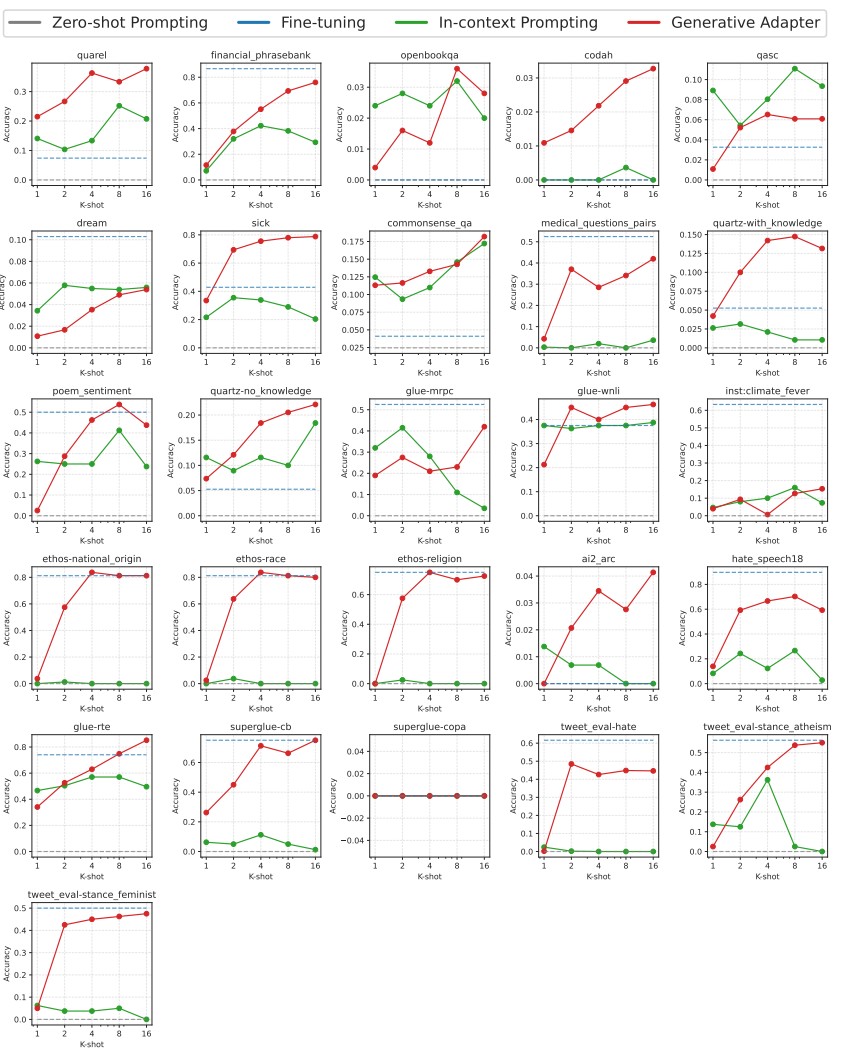

Figure 6: In-context learning evaluation of Generative Adapter, based on Mistral-7B-Instruct, across 26 test datasets from MetaICL.

| Model | Dataset | Methods | F1 | | | | | | |
|---|---|---|---|---|---|---|---|---|---|
| | | | 512 | 1K | 2K | 4K | 8K | 16K | 32K |
| Mistral | SQuAD | Zero-Shot Prompting | 10.8 | | | | | | |
| | | Supervised Fine-tuning | 20.7 | | | | | | |
| | | Continuous Pretraining | 30.0 | | | | | | |
| | | In-context Prompting | 45.4 | 44.9 | 43.6 | 42.6 | 42.5 | 38.6 | 35.1 |
| | | GenerativeAdapter | 48.8 | 43.0 | 39.9 | 35.9 | 33.8 | 30.3 | 28.0 |
| | StreamingQA | Zero-Shot Prompting | 13.6 | | | | | | |
| | | Supervised Fine-tuning | 19.5 | | | | | | |
| | | Continuous Pretraining | 22.2 | | | | | | |
| | | In-context Prompting | 47.2 | 48.7 | 48.1 | 48.7 | 48.0 | 46.0 | 39.3 |
| | | GenerativeAdapter | 51.5 | 49.3 | 44.7 | 40.9 | 36.7 | 32.7 | 32.0 |
| LLama2 | SQuAD | Zero-Shot Prompting | 14.6 | | | | | | |
| | | Supervised Fine-tuning | 20.7 | | | | | | |
| | | Continuous Pretraining | 23.9 | | | | | | |
| | | In-context Prompting | 64.8 | 60.6 | 55.4 | 44.9 | 25.2 | 9.6 | 6.4 |
| | | GenerativeAdapter | 36.2 | 32.5 | 31.0 | 28.9 | 28.2 | 24.9 | 23.6 |
| | StreamingQA | Zero-Shot Prompting | 18.0 | | | | | | |
| | | Supervised Fine-tuning | 18.9 | | | | | | |
| | | Continuous Pretraining | 20.5 | | | | | | |
| | | In-context Prompting | 61.2 | 61.3 | 58.0 | 46.8 | 27.8 | 17.5 | 11.6 |
| | | GenerativeAdapter | 42.9 | 37.8 | 34.4 | 32.6 | 28.7 | 26.0 | 25.7 |

Table 5: All results of the QA accuracy on SQuAD and StreamingQA.

---

**Prompting for Document-based Question Answering**

{Context}

## Instruction: Answer the question based on the context above. Respond with a short phrase only. Keep the answer short and concise, without any explanation or additional words

Question: {Question}
Answer:

---

**Prompting for MetaICL**

Input: {demo input}
Output: {demo output}
{ . . . k-shot demonstrations . . . }

## Instruction: Based on the demonstration above, provide a short and concise answer, without any explanation or additional words.

Input: {input}
Output:

---

Figure 7: Prompts used in the document-based QA and in-context learning evaluation.

Table 6: The F1 scores (along with the number of gist tokens in parentheses) on the StreamingQA dataset to show complementary benefits on top of RAG.

| Context Length | 512 | 1K | 2K | 4K | 8K | 16K | 32K |
|---|---|---|---|---|---|---|---|
| GenerativeAdapter | 51.5 (0K) | 49.3 (0K) | 44.7 (0K) | 40.9 (0K) | 36.7 (0K) | 32.7 (0K) | 32.0 (0K) |
| GenerativeAdapter + Context | 67.8 (0.5K) | 67.8 (1K) | 61.1 (2K) | / | / | / | / |
| GenerativeAdapter + RAG | 61.9 (0.1K) | 63.6 (0.1K) | 60.8 (0.1K) | 60.9 (0.1K) | 60.1 (0.1K) | 59.1 (0.1K) | 56.5 (0.1K) |

Table 7: The F1 scores (along with the number of gist tokens in parentheses) on the StreamingQA dataset for Ultragist with different compression ratios.

| Context Length | 512 | 1K | 2K | 4K | 8K | 16K | 32K |
|---|---|---|---|---|---|---|---|
| Ultragist (Compression Ratio=2) | 63.5 (0.3K) | 63.6 (0.5K) | 62.3 (1K) | 61.9 (2K) | 61.8 (4K) | 62.1 (8K) | 51.0 (16K) |
| Ultragist (Compression Ratio=8) | 57.6 (0.1K) | 55.7 (0.1K) | 55.4 (0.3K) | 55.7 (0.5K) | 54.0 (1K) | 53.0 (2K) | 51.1 (4K) |
| Ultragist (Compression Ratio=32) | 32.5 (0.1K) | 31.1 (0.1K) | 30.1 (0.1K) | 32.8 (0.1K) | 33.0 (0.3K) | 32.0 (0.5K) | 31.4 (1K) |
| GenerativeAdapter | 51.5 (0K) | 49.3 (0K) | 44.7 (0K) | 40.9 (0K) | 36.7 (0K) | 32.7 (0K) | 32.0 (0K) |

Table 8: Comparison between Generative Adapter and Ultragist on MetaICL.

| Method | Classification | Non-classification |
|---|---|---|
| Ultragist (256 tokens) | 41.1 | 7.5 |
| In-context prompting | 60.5 | 6.7 |
| Finetune | 71.8 | 10.5 |
| GenerativeAdapter | 63.7 | 14.9 |

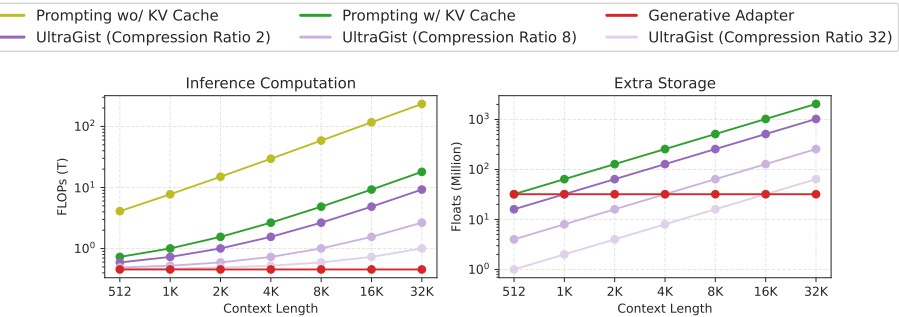

Figure 8: Inference computation and storage requirements for Generative Adapter and baseline methods on StreamingQA.

