# OpenReview forum: "Generative Adapter: Contextualizing Language Models in Parameters with A Single Forward Pass"
_ICLR.cc/2025/Conference — ICLR 2025 Poster_

### Official Review · Reviewer_PYAH · 2024-10-31

**Soundness:** 4
**Presentation:** 4
**Contribution:** 3
**Rating:** 6
**Confidence:** 4

**Summary:**

This paper proposes a novel method to integrate the information into the parameters. The experimental results are comprehensive and demonstrate the effectiveness of this method.

**Strengths:**

1. The topic is important.
2. The idea of this paper is novel and clean, inspired by fast weights.
3. The experimental results show the effectiveness of this method. it is impressive to see that model is able to inject 32k tokens into the parameters.

**Weaknesses:**

1. **Missing related works**: GenerativeAdaptor adds the adaptor which is quite similar to memory-based methods, where people would add a memory module on the large language models [1,2]. Perhaps the paper should discuss some related papers and the correlations between them.

2. **Size of the additional parameters**: As stated in line 264, GenerativeAdaptor introduces 500 million parameters, however, even using 500 million parameters cannot achieve comparative performance as In-context Prompting (as shown in Figure 2(a)). Consider saving all the tokens in the context, which leads to 32768 * 4096 = 134 million parameters. This is fine, as the proposed method has smaller computation cost than storing all the previous tokens, but this might be something to consider, as it shows that the parameters may not be fully utilized, or the information in the parameters might be overly sparse.

[1] Memllm: Finetuning llms to use an explicit read-write memory
[2] MemoryLLM: Towards Self-Updatable Large Language Models

**Questions:**

I don't have any questions.

---

> ### Author Response · Authors · 2024-11-20
> **Response to Reviewer PYAH**
>
> Thank you for recognizing the importance of the problem and the novelty of our idea. We addressed the weaknesses and questions you raised as follows.
>
> ---
>
> > Q1: GenerativeAdaptor introduces 500 million parameters, however, even using 500 million parameters cannot achieve comparative performance as In-context Prompting (as shown in Figure 2(a)). Consider saving all the tokens in the context, which leads to 32768 * 4096 = 134 million parameters.
>
> We want to provide further clarification on the parameter sizes of the adapter generator ($G$) vs the generated adapter ($\Delta W$) applied on top of the backbone LM for inference. The adapter generator has approximately 500 million parameters and is independent of the context, whereas the generated adapter is only about 32 million parameters across all experiments.
>
> If we aim to retain information for multiple users, we would only need to store the generated adapter for each user, and a single copy of the adapter generator (along with the language model). In contrast, maintaining the token hidden states or KV cache for the entire context, as suggested, would result in significant storage overhead. We believe this distinction highlights the efficiency of our approach in scenarios requiring scalable storage solutions.
>
> ---
> > Q2: GenerativeAdaptor adds the adaptor which is quite similar to memory-based methods, where people would add a memory module on the large language models [1,2].
>
> Thank you for pointing out this connection. We acknowledge that memory-based methods share some similarities with our approach. However, our work focuses on preserving the standard transformer-based language model architecture while proposing a method to efficiently and effectively update model parameters based on the given context. We will ensure to discuss and cite relevant works in the future, including the ones you mentioned.
>
> ---
> Thank you once again for your thoughtful questions and valuable feedback! Please don’t hesitate to let us know at your earliest convenience if you have any further questions or concerns, or if there are additional experiments you would like us to conduct.

---

> > ### Comment · Reviewer_PYAH · 2024-11-23
> > **Response to the authors**
> >
> > Thank the authors for addressing my concerns.
> >
> > [Q1] I think it makes sense.
> > [Q2] I don't think the current work is **updating model parameters** as the current model is compressing the information into the adaptor. As said in [Q1], the context is compressed into 32 million parameters. As Fast Weights has super close relationship with Memory-based RNNs, I think this work is also closely related to memory-based methods.

---

> > > ### Author Response · Authors · 2024-11-23
> > > **Response to Reviewer PYAH**
> > >
> > > Thank you for your follow-up regarding Q2.
> > >
> > > **What We Mean by "Updating Model Parameters"**
> > >
> > > Unlike traditional updates using gradient descent, which require multiple training steps to adjust parameters, our approach generates a context-dependent additive adapter (e.g., a LoRA) in a single forward pass. This adapter is merged directly into the Transformer weights, enabling parameter updates based on the provided context without iterative optimization. Compared to memory-based methods, our approach eliminates the need for additional memory components during inference, focusing instead on efficient and effective parameter updates without relying on external storage.
> > >
> > > Below is an abstract of how the adapter is generated based on a given context (detailed in Section 2.2). For each linear projection layer in the $ l $-th Transformer block, the adapted weights are parameterized as:  $ W^{(l)} = W^{(l)}\_{base} + W^{(l)}\_{\Delta}, $ where $ W^{(l)}\_{{base}} $ comes from the base model, and $ W^{(l)}\_{\Delta} $ is generated based on the given context $ C $ (Line 160–165). The process involves encoding $ C $ into hidden states $ H^{(l)} $, which are then used to compute $ W^{(l)}\_{\Delta} $ via an outer product and normalization (Line 175-182). This context-dependent $ W^{(l)}\_{\Delta} $ is directly integrated into the model.
> > >
> > > **Relationship with Memory-Based Methods**
> > >
> > > We acknowledge the connection between our approach and memory-based methods, such as MemLLM [1] and MemoryLLM [2]. All methods aim to compress contextual or historical information into a knowledge representation that language models can utilize more effectively. All works include additional designs to generate the representations: MemLLM and MemoryLLM [1, 2] use tuned language models to generate relation triplets or dense representations, whereas GenerativeAdapter uses an adapter generator to produce adapters.
> > >
> > > However, after the memory update, memory-based methods rely on the explicit storage of external knowledge (e.g., relation triplets or dense representations). In contrast, in our approach, the updated LM generates outputs to user inputs entirely within the Transformer architecture during inference. When new context arrives, the adapter generator creates an adapter, which is directly merged into the language model.
> > >
> > > Additionally, memory-based RNNs like LSTMs and GRUs are designed to retain long-term dependencies within their architecture. In contrast, our work is tailored to large Transformer-based models, focusing on efficient parameter updates by a given context.
> > >
> > > We will cite these related works and discuss their connections in the final version of the paper. Thank you for highlighting this point and for your thoughtful feedback.
> > >
> > > [1] MemLLM: Finetuning LLMs to Use an Explicit Read-Write Memory
> > > [2] MemoryLLM: Towards Self-Updatable Large Language Models
> > >
> > > ---
> > >
> > > Please do not hesitate to let us know if you have any further questions or need additional clarification. We also kindly request your reevaluation of our paper. Thank you for your time and thoughtful feedback!

---

> > > ### Author Response · Authors · 2024-11-25
> > >
> > > If the updates and clarifications we have provided address your concerns and improve the quality of the work to your satisfaction, we kindly ask you to consider reflecting this in your overall assessment. We greatly appreciate your feedback and efforts in reviewing our submission.

---

> > > ### Author Response · Authors · 2024-12-02
> > >
> > > Thank you so much for your time and valuable feedback. We would like to kindly remind you that Dec 2nd is the final day for reviewers to post messages to the authors. If you have any additional questions or concerns, please let us know at your earliest convenience.
> > > If you feel that the changes and clarifications made have sufficiently addressed your queries and improved the quality of the work, we would greatly appreciate it if you could consider this in your overall evaluation.

---

### Official Review · Reviewer_mTD9 · 2024-11-04

**Soundness:** 2
**Presentation:** 3
**Contribution:** 3
**Rating:** 6
**Confidence:** 3

**Summary:**

The paper proposes GenerativeAdapter, which is a method to introduce unseen knowledge to LMs without finetuning cost or extra prompts. It generates a new adapter for the target unseen knowledge and utilize it to adapt the LMs.

**Strengths:**

- The paper addresses the critical challenge of adapting LMs to incorporate new knowledge effectively, a fundamental issue since LMs can only utilize the information they were originally trained on.
- It introduces an interesting approach of training another generator to generate an adaptor and optimize adaptation cost

**Weaknesses:**

- Some of the important experiments are either missing or have questionable setting
    - It is suspicious that the the settings for experiment 4.1 are fair for all of the baselines. It seems like the knowledge that each baseline has seen are different.
    - In section 5, the paper mentions the preliminary study shows the quality of the adapter generator is highly correlated with the final evaluation metrics. It would have been better if the results that support this claim is included in the paper. Many times, the training objective is not exactly aligned with the downstream task.
    - It would be better if the experiment to see how the adaptor affects the original ability of the LMs is included in the paper.

**Questions:**

- Is there a suspected reason why the unnormalized adaptor contains extreme values?
- In the formulations in section 2.3, there is a  G(x_{1:m}). In this case, shouldn't the hidden states input to the generator, not the input tokens?
-  In the completion loss formulation, shouldn't the output just x_{m+1}, not x{m+1},...,x_n?

---

> ### Author Response · Authors · 2024-11-20
> **Response to Reviewer mTD9**
>
> Thank you for recognizing the strengths of our work\. We address the identified weaknesses and your questions below.
>
> ---
> > Q1: It seems like the knowledge that each baseline has seen are different in experiment 4.1.
>
> Closed-book prompting and supervised finetuning are two baselines that do not receive any context during test time. The continual pretraining baseline sees all documents (complete knowledge) during the training stage. In contrast, only a subset of documents (partial knowledge) used for inference is seen by both in-context prompting and our method. In our experiments, we use the same subset of documents for both in-context prompting and our method.
> Thus, **our generative adapter does not have any advantage in terms of access to more documents (knowledge) compared to continual pretraining and in-context prompting**. We hope this clears up any misunderstandings regarding the fairness of our comparisons.
>
> ---
> > Q2: the paper mentions … quality of the adapter generator is highly correlated with the final evaluation metrics.  It would have been better if the results that support this claim is included in the paper.
>
>
> We provide additional evaluation results in the table below. Specifically, we compare two normalization methods: SVD normalization and Frobenius normalization, and evaluate them using reconstruction perplexity, completion perplexity, and F1 score on the MSC dataset (as used in Section 4.3).
>
> Our findings indicate that **reconstruction perplexity is strongly correlated with fact recall, as measured by the F1 score**. For example, the SVD normalization method achieves a much lower reconstruction perplexity and a significantly higher F1 score compared to Frobenius normalization.
>
> | Setting           | Reconstruction PPL | Completion PPL | MSC F1 |
> |------------------------|------------------------|---------------------|------------|
> | Default (SVD Norm)     | 1.75                  | 7.40               | 40.2       |
> | Frobenius Norm         | 7.72                  | 7.32               | 27.1       |
>
>
> ---
> > Q3: Is there a suspected reason why the unnormalized adaptor contains extreme values?
>
> We suspect this behavior arises from the distribution of hidden states. Specifically, the hidden states of tokens are not uniformly distributed as a Gaussian but exhibit higher frequencies in certain directions. Consequently, the sum of the outer products of these hidden states results in a significant eigenvalue in the dominant direction, leading to instability.
>
> ---
> > Q4: In the formulations in section 2.3, there is a G(x_{1:m}). In this case, shouldn't the hidden states input to the generator, not the input tokens?
>
> Here, the intention is to describe that the adapter is generated based on the first m tokens and is subsequently used to compute the log-likelihood on the updated model. We will clarify this point in the future version of the paper.
>
> ---
> > Q5: In the completion loss formulation, shouldn't the output just $x_{m+1}$, not $x_{m+1},...,x_n$?
>
> The rationale here is that the adapter is generated based on $x_1$ to $x_m$​, and the log-likelihood is then computed over $x_{m+1},…,x_n​$ using the updated model. This reflects the model's ability to predict the continuation of the sequence after being contextualized with the initial context.
>
> ---
> Thank you once again for your thoughtful questions and valuable feedback! Please don’t hesitate to let us know at your earliest convenience if you have any further questions, concerns, or if there are additional experiments you would like us to conduct.

---

> ### Author Response · Authors · 2024-11-23
>
> Please don’t hesitate to let us know if you have any additional questions or need further clarification. We’d also love to hear if we’ve addressed all your concerns. Thank you again for your thoughtful feedback!

---

> ### Author Response · Authors · 2024-11-25
>
> If the updates and clarifications we have provided address your concerns and improve the quality of the work to your satisfaction, we kindly ask you to consider reflecting this in your overall assessment. We greatly appreciate your feedback and efforts in reviewing our submission.

---

> > ### Comment · Reviewer_mTD9 · 2024-11-28
> > **Thank you for your response!**
> >
> > Most of my concerns are addressed. However, I have 2 follow-up questions.
> >
> > 1. Q1: You said only partial knowledge was used for your method, but since your method can utilize many context chunks, can't you utilize complete knowledge by chunking them?
> > 2. I believe the experiment to see how the adaptor affects the original ability of the LMs is still missing
> >
> > Additionally, I think the contribution and the motivation the paper claims is a little insufficient to increase the score

---

> > > ### Author Response · Authors · 2024-12-02
> > >
> > > Thank you for your questions.
> > >
> > > > Q1: You said only partial knowledge was used for your method, but since your method can utilize many context chunks, can't you utilize complete knowledge by chunking them?
> > >
> > > We would like to clarify that the primary goal of GenerativeAdapter is to efficiently adapt pretrained LMs on the fly. Hence, GenerativeAdapter is designed to adapt at a finer granularity such as document chunks during streaming, rather than adapting for a whole corpus with thousands of documents.
> > >
> > > On the other hand, prior research has shown that the continual pretraining approach struggles with low-frequency facts, especially when the information appears only once in the data [1]. As a result, it has to rely on the entire corpus to be effective, making it less flexible for adaptation at a finer granularity.
> > >
> > > In our experiments, for each question, we use relevant documents to generate the corresponding adapter, rather than creating a single adapter for the entire corpus (referred to as "complete knowledge"). Our results show that GenerativeAdapter does not require the whole corpus to achieve satisfactory accuracy, providing greater flexibility for adaptation at a finer granularity (e.g., using just one or two document chunks). Additionally, the same adapter can be stored and reused for multiple questions, as long as the relevant chunks contain the necessary answer information.
> > >
> > > [1] Zhu et al. (2023) Physics of Language Models: Part 3.1, Knowledge Storage and Extraction
> > >
> > > ---
> > > > I believe the experiment to see how the adaptor affects the original ability of the LMs is still missing.
> > >
> > > We acknowledge the importance of understanding how the adapter impacts the original abilities of the language model. However, because the base models (Mistral and Llama) are instruction-tuned on private datasets, whereas GenerativeAdapter is tuned using different datasets, a direct comparison is not feasible. Nevertheless, we conducted the experiment described below to demonstrate that the adapted model still retains knowledge memorized during pretraining.
> > >
> > > We evaluate the base and adapted models on the MMLU dataset, which measures knowledge memorization across 57 subjects in STEM and other domains. Using Mistral-7B-Instruct-v0.2 as the base model, we compare its performance with and without a generative adapter. For the adapted model, we generate an adapter using a system prompt as context: "You are a helpful AI assistant."
> > >
> > > | Model                               | MMLU Accuracy (0-shot)  |
> > > |----------|----------|
> > > | Base Model (Mistral-7B-Instruct-v0.2) | 0.574    |
> > > | Base Model with Generative Adapter  | 0.576    |
> > >
> > > The almost identical accuracy between the base and adapted models suggests the adapter effectively incorporates new instructions without harming the model's original abilities.
> > >
> > > ---
> > > To conclude, we want to highlight these contributions in our paper:
> > > - Generative Adapter Framework: We propose a novel model adaptation framework, GenerativeAdapter, which processes (a stream of) input contexts in a single forward pass to generate an adapter. This adapter directly merges with the base language model parameters.
> > > - Effective Knowledge Integration: Generative Adapter effectively integrates factual knowledge into language model weights on a single document. In contrast, conventional continual pretraining requires training on a large corpus, often needing facts to be repeated multiple times for memorization.
> > > - Effective In-Context Learning: Generative Adapter outperforms prompting in in-context learning accuracy and offers more efficient inference once the LM is contextualized with examples.

---

### Official Review · Reviewer_apSm · 2024-11-05

**Soundness:** 3
**Presentation:** 2
**Contribution:** 3
**Rating:** 6
**Confidence:** 2

**Summary:**

The paper introduces GenerativeAdapter, a method for adapting pretrained large language models (LLMs) using lightweight, parameter-efficient adapters generated by a self-supervised generator network. This approach aims to enhance LLM adaptability without incurring the computational cost of fine-tuning or the inference overhead of prompting. Evaluations are conducted across three main areas: factual knowledge acquisition, learning from demonstrations, and user personalization, showing that GenerativeAdapter is effective in reducing computational load and increasing model adaptability to various contexts.

**Strengths:**

1. The paper presents an efficient approach to adapting LLMs in real-time by using a lightweight, bilinear adapter generator. This mechanism effectively reduces computational costs associated with full model fine-tuning and inference-time overhead due to prompting, which is particularly advantageous for applications on edge devices.

2. The authors evaluate *GenerativeAdapter* across diverse, practical contexts such as document-based knowledge acquisition, in-context learning, and user personalization. These evaluations, especially in a personalized setting, demonstrate the adaptability of the method and its effectiveness in various contexts that align with real-world demands.

**Weaknesses:**

1. The authors provide detailed descriptions of the model architecture, adapter generation mechanism, and datasets. However, hyperparameter settings and implementation details could be expanded, especially regarding stability mechanisms like SVD normalization, which could impact reproducibility. It is unclear if the code or models will be open-sourced for verification.
2. The analysis of *GenerativeAdapter*’s performance in different scenarios is comprehensive; however, the lack of prompt  engineering effort for baselines could introduce an unfair comparison. A clarification on the extent of prompt engineering would ensure a fairer assessment.
3. While *GenerativeAdapter* achieves superior performance over specific context lengths and adaptation scenarios, claims of general "efficiency" for "personalization" are strong, given the specific constraints of the study. The authors should emphasize that observed benefits apply to controlled experiments rather than broad generalizations.
4. The paper introduces concepts like "test-time contextualization" and "parameter-efficient" without specific definitions, which could hinder comprehension for readers less familiar with these terms. Explicit definitions or brief explanations in the methodology section would help maintain clarity.

**Questions:**

How does GenerativeAdapter perform in scenarios where the context information is dynamic or frequently changing, such as real-time conversational agents? It would be interesting to see if the method’s efficiency and adaptability hold in continuously evolving contexts.

---

> ### Author Response · Authors · 2024-11-20
> **Response to Reviewer apSm**
>
> Thank you for recognizing the effectiveness and efficiency of our work in adaptation. We provide our responses to the weaknesses and questions you highlighted in detail below.
>
> ---
>
> > Q1: hyperparameter settings and implementation details could be expanded, especially regarding stability mechanisms like SVD normalization, which could impact reproducibility.
>
> We have included the hyperparameter settings and implementation details in Appendix B. Regarding SVD normalization, we use the `torch.svd_lowrank()` function from the PyTorch library, as described in the "Implementation" paragraph. We agree on the importance of reproducibility and will improve clarity and provide additional details in the next version of the paper.
>
> ---
>
> > Q2: the lack of prompt engineering effort for baselines could introduce an unfair comparison.
>
> The prompt templates used in this paper are listed in Figure 7 in the appendix. Since both backbone LMs are instruction-tuned, we observed them to be quite reliable in performing the standard tasks considered in this paper. While prompt engineering could potentially benefit both our model and the baselines, we leave this for future exploration.
>
> ---
>
> > Q3: The paper introduces concepts like "test-time contextualization" and "parameter-efficient" without specific definitions.
>
> We introduce "test-time contextualization" in Section 2.1 (Line 135), where it refers to scenarios where context arrives incrementally as a stream of data (e.g., a continuous flow of documents or dialogue sessions). The goal is to obtain an updated model capable of responding to user instructions using the information provided in the context C
>
> For parameter-efficient fine-tuning (PEFT), it involves updating only a small subset of parameters instead of the entire language model. We discuss related works on PEFT in Section 6 (Line 516).
>
> ---
> > Q4: claims of general "efficiency" for "personalization" are strong, given the specific constraints of the study.
>
> In this paper, we define personalization as the ability of language models to analyze users’ behavior and memorize their preferences, which is key to creating a tailored and engaging user experience (Line 412). Specifically, we evaluate the model’s ability to memorize user information in conversational contexts. We will carefully revise the wording in future versions to ensure that our claims align with the specific scope of our study.
>
> ---
> > Q5: How does GenerativeAdapter perform in scenarios where the context information is dynamic or frequently changing, such as real-time conversational agents? It would be interesting to see if the method’s efficiency and adaptability hold in continuously evolving contexts.
>
> Our experiments on document-based QA are designed to address scenarios with dynamic context. For instance, when the context length extends to 32K tokens, we dynamically split the context into 1K chunks and update the generative adapter incrementally, as described in Section 2.2. This approach demonstrates our method’s ability to handle evolving contexts efficiently. We appreciate your suggestion to further explore real-time conversational agents. However, as we are not aware of a suitable dataset for such scenarios, we will consider this as a direction for future work.
>
> ---
> Thank you once again for your thoughtful questions and valuable feedback! Please don’t hesitate to let us know at your earliest convenience if you have any further questions, concerns, or if there are additional experiments you would like us to conduct.

---

> ### Author Response · Authors · 2024-11-23
>
> Please don’t hesitate to let us know if you have any additional questions or need further clarification. We’d also love to hear if we’ve addressed all your concerns. Thank you again for your thoughtful feedback!

---

> ### Author Response · Authors · 2024-11-25
>
> If the updates and clarifications we have provided address your concerns and improve the quality of the work to your satisfaction, we kindly ask you to consider reflecting this in your overall assessment. We greatly appreciate your feedback and efforts in reviewing our submission.

---

### Official Review · Reviewer_1mHi · 2024-11-05

**Soundness:** 3
**Presentation:** 4
**Contribution:** 2
**Rating:** 5
**Confidence:** 4

**Summary:**

This paper proposes a neural module called generative adapter, which once trained can accept a stream of input contexts to adapt transformers on the fly. The paper shows that generative adapters work well at lower computational cost than in-context learning on several NLP tasks.

**Strengths:**

- The method is simple and well-motivated.
- The method works nicely in a streaming setting to continually update weights using lower number of FLOPs due to being able to reuse weights computed at the previous time step of the stream.
- The paper is clearly written and easy to follow.

**Weaknesses:**

- I think the Ultragist baseline should be reported in $\S 4.1$ and  $\S 4.2$, not just $\S 4.3$. Gisting is a natural comparison and I am glad to see it compared against in some form, but I think it could be reported more widely in the paper.
- Using llama-2-7b doesn't provide much value in my opinion. The 4k context length makes Figure 2b hard to reason about, and I don't see why all llama-2-7b experiments couldn't be done with llama-3.1-8b instead for similar cost.

**Questions:**

- From what I understand, generative adapters are Hypernetworks [[Ha et al](https://arxiv.org/abs/1609.09106)], but that hasn't been discussed anywhere. Do the authors agree with this assessment, and if so, how would they contextualize similar work using hypernetworks for continual learning [[Oswald et al](https://arxiv.org/abs/1906.00695), [Vladymyrov et al](https://arxiv.org/abs/2301.04584)]?
- Why is the continual pretraining baseline so much lower than the generative adapter baseline? This sticks out and is surprising to me, and makes me think it is poorly tuned.
- I think a valuable baseline (especially since continual pretraining seems to underperform) might be using the generative adapter loss function (Eqn 5) but optimize the weights of the model directly instead of the generative adapter. If I understood correctly, I don't see any reason why this shouldn't be possible, and if generative adapter is in the ballpark of this baseline, it's a positive signal. Do the authors agree with this assessment?

---

> ### Author Response · Authors · 2024-11-20
> **Response to Reviewer 1mHi (part 1)**
>
> We appreciate your valuable questions and suggestions! Below, we provide responses to your questions and address the identified weaknesses.
>
> ---
> > Q1: Why is the continual pretraining baseline so much lower than the generative adapter baseline?
>
> First, **we would like to emphasize that vanilla continual pretraining has been shown to be less effective for enabling LMs to acquire knowledge for downstream tasks**, as reported in the literature [1, 3]. This limitation motivates various efforts, including our work, to address this gap. Two hypotheses have been proposed to explain this phenomenon:
>
> 1. The unweighted next-token prediction loss is not optimal for pretrained LMs [1].
> 2. Additional (e.g., synthesized) data about each fact is necessary [3, 4].
>
>
> Second, **our continual pretraining baseline follows the setups from prior studies [1, 2], with sufficient tuning**. QA accuracy results on SQuAD and StreamingQA align well with those reported in earlier work. Below are the results for comparison:
>
> | Base Model                  | StreamingQA | SQuAD |
> |-----------------------------|-------------|-------|
> | GPT-2 XL ([1])              | ~12.00      | /     |
> | Llama2-7B ([2])             | 13.54       | 17.01 |
> | Mistral-7B-Instruct (Ours)  | 30.0        | 22.2  |
> | Llama2-7B-Chat (Ours)       | 23.9        | 20.5  |
>
> * [1] Hu et al. (2023) Meta-Learning Online Adaptation of Language Models
> * [2] Tack et al. (2024) Online Adaptation of Language Models with a Memory of Amortized Contexts
> * [3] Zhu et al. (2023) Physics of Language Models: Part 3.1, Knowledge Storage and Extraction
> * [4] Yang et al. (2024) Synthetic continued pretraining
>
>
> ---
> > Q2: Ultragist should be reported in sections 4.1 and 4.2.
>
> While Ultragist can be compared, we excluded it from sections 4.1 and 4.2 for the following reasons:
> 1. **Ultragist is a lossy version of in-context prompting**. As a context compression method, Ultragist can be seen as a lossy version of in-context prompting, which represents the upper bound of performance. Our primary focus is to show whether our method can close the gap with in-context prompting. To address the reviewer’s concerns, we evaluated Ultragist (with a compression limit of 256 tokens) on the MetaICL dataset, and the results are reported below. The findings show that Ultragist performs worse than the in-context prompting baseline.
> | Method                | Classification | Non-classification |
> |-----------------------|----------------|---------------------|
> | Ultragist (256 tokens) | 41.1             | 7.5                  |
> | In-context prompting | 60.5           | 6.7                 |
> | Finetune            | 71.8           | 10.5                |
> | Generative Adapter  | 63.7           | 14.9                |
>
>
> 2. **Ultragist's performance depends on the compression ratio.** In Section 4.3, we tested Ultragist across a range of compression ratios, from 256 tokens to 2K tokens. In contrast, Sections 4.1 and 4.2 focus on evaluations involving varying context lengths (Section 4.1) and numbers of examples (Section 4.2), making a fair comparison challenging.
>
>
>
> ---
> > Q3: llama-2-7B doesn’t provide much value and llama-2-7B should be replaced by llama-3.1-8b.
>
> We chose Llama-2 because it was one of the most widely used open-source LLMs at the time of submission. Its differences from Mistral—such as being developed by a different company with distinct pretraining strategies (data, vocabulary, and context window size)—make it ideal for testing the robustness of our method. The Llama3.1 model family, released in July 2024, was too close to our submission deadline for inclusion.
>
> **Our results on Mistral-7B and Llama2-7B effectively demonstrate the strength of generative adapters.** On both models, generative adapters outperformed continual pretraining in knowledge acquisition efficiency, achieving better results with shorter contextualization times. Furthermore, the generative adapter significantly narrowed the performance gap between continual pretraining and in-context prompting on Mistral-7B for sequences up to 32K tokens.
>
> To address your concern, we provide below the results of in-context prompting on Llama3.1-8B-Instruct using the StreamingQA dataset. These results show a similar trend to Mistral-7B-Instruct. We speculate that comparisons between generative adapters and other baselines on Llama3.1 would align with those observed on Mistral.
>
> | Context Length          | 512  | 1k   | 2k   | 4k   | 8k   | 16k  |
> |--------------------------|------|------|------|------|------|------|
> | Llama3.1-8B-Instruct    | 70.0 | 70.3 | 69.9 | 69.4 | 68.1 | 68.8 |
> | Mistral-7B-Instruct     | 47.0 | 47.2 | 47.5 | 47.4 | 46.5 | 45.0 |

---

> ### Author Response · Authors · 2024-11-20
> **Response to Reviewer 1mHi (part 2)**
>
> > Q4: I think a valuable baseline (especially since continual pretraining seems to underperform) might be using the generative adapter loss function (Eqn 5) but optimize the weights of the model directly instead of the generative adapter.
>
> As mentioned in our responses to Q1, continual pretraining with next-token prediction is a common approach for domain adaptation but is not very effective for injecting new knowledge into model parameters.
>
> The loss function in Eqn 5, designed for training the adapter generator by mapping text sequences to adapter matrices, combines completion and reconstruction components. Using it directly for continual training could cause model collapse, e.g., the LM simply repeats the first half of the input. However, your suggestion points to an important research direction: developing better training objectives for knowledge acquisition, which could bring significant improvements.
>
> ---
>
> > Q5: how would they contextualize similar work using hypernetworks for continual learning
>
> Thank you for pointing out these Hypernetwork papers. While they are related to fast weights, they differ from the fast weights concept discussed in our paper. Due to space constraints, we excluded them from the initial version.
>
> Hypernetworks, such as those in [1][2], are variants of [3], which we cited. However, these methods train the hypernetwork and backbone jointly, unlike our approach. They also focus on specific tasks, whereas we aim to enhance frozen pretrained LMs for general language tasks. Our adapter network, once trained, supports diverse scenarios with varying context characteristics.
>
> Most hypernetwork-based methods [1][2] target supervised improvements in in-context learning but do not address recalling document knowledge or user facts for tasks like question answering.
>
> We will include this discussion in the final version.
>
> * [1] Vladymyrov et al. (2024) Continual HyperTransformer: A Meta-Learner for Continual Few-Shot Learning
> * [2] Oswald et al. (2022) Continual Learning with Hypernetworks
> * [3] Ba et al. (2016) Using Fast Weights to Attend to the Recent Past
>
> ---
> Thank you once again for your thoughtful questions and valuable feedback! Please don’t hesitate to let us know at your earliest convenience if you have any further questions or concerns, or if there are additional experiments you would like us to conduct.

---

> ### Author Response · Authors · 2024-11-23
>
> Please feel free to let us know if you have any additional questions or need further clarification. We’re also curious to know if we’ve addressed all your concerns. Thank you again for your valuable feedback!

---

> > ### Comment · Reviewer_1mHi · 2024-11-23
> > **Response**
> >
> > Thanks for your response. I think most of my comments have been adequately addressed. However I don’t buy that few shot prompting being an upper bound for gisting means gisting is not a valuable baseline. It seems like few-shot prompting would also then be an upper bound for Generative Adapter. At the end of the day Generative adapter falls into the category of methods that do few shot learning without prompting (with the added streaming benefits), so without adequate comparisons to baselines in the same category of methods, it’s hard for me to argue for acceptance.

---

> > > ### Author Response · Authors · 2024-11-25
> > >
> > > Thank you for your question. We have provided additional experimental results for Ultragist in both Sections 4.1 and 4.2 below.
> > >
> > > **Additional Experimental Results (Section 4.1)**
> > >
> > > The results below show the F1 scores (along with the number of gist tokens in parentheses) on the StreamingQA dataset for Ultragist with different compression ratios.
> > >
> > > | Context Length                | 512          | 1K           | 2K           | 4K           | 8K           | 16K          | 32K          |
> > > |----------|------|------|------|------|------|------|------|
> > > | Ultragist (Compression Ratio=2)  | 63.5 (0.3K)  | 63.6 (0.5K)  | 62.3 (1K)    | 61.9 (2K)    | 61.8 (4K)    | 62.1 (8K)    | 51.0 (16K)   |
> > > | Ultragist (Compression Ratio=8)  | 57.6 (0.1K)  | 55.7 (0.1K)  | 55.4 (0.3K)  | 55.7 (0.5K)  | 54.0 (1K)    | 53.0 (2K)    | 51.1 (4K)    |
> > > | Ultragist (Compression Ratio=32) | 32.5 (0.1K)  | 31.1 (0.1K)  | 30.1 (0.1K)  | 32.8 (0.1K)  | 33.0 (0.3K)  | 32.0 (0.5K)  | 31.4 (1K)    |
> > > | GenerativeAdapter                | 51.5 (0K)    | 49.3 (0K)    | 44.7 (0K)    | 40.9 (0K)    | 36.7 (0K)    | 32.7 (0K)    | 32.0 (0K)    |
> > >
> > > As expected, Ultragist’s performance drops significantly as the compression ratio increases.
> > >
> > > We acknowledge that both the Ultragist and our method involve compressing context into dense representations. Thus, we would like to highlight several advantages of GenerativeAdapter due to the fundamental differences in methodology.
> > >
> > > **Inference Efficiency**: Since GenerativeAdapter directly modifies the model parameters, it has the same inference time as the base model (after contextualization). In contrast, Ultragist’s inference and storage costs (after compression) depend on the total number of gist tokens, which is approximately equal to the context length divided by the compression ratio. Given the inference efficiency of GenerativeAdapter, it is more suitable for scenarios where the model is contextualized once and reused multiple times during inference, making it particularly efficient for resource-constrained environments like edge computing. To provide a clearer comparison, we have included a new figure below alongside Figure 3 in the paper, showing Ultragist’s inference computation and extra storage costs.
> > >
> > > New Figure: [![Figure]https://i.postimg.cc/NF8dt4zW/results-efficiency-ultragist.jpg)](https://i.postimg.cc/NF8dt4zW/results-efficiency-ultragist.jpg)
> > >
> > > **Complementary Benefits**: Furthermore, our GenerativeAdapter can be combined with prompting or other prompt compression approaches to provide complementary benefits. To demonstrate this, we combined GenerativeAdapter with Retrieval-Augmented Generation (RAG) as an example. In this hybrid approach, we keep the contextualization phase unchanged and prepend the most relevant 100-token chunk (retrieved via BM25) to the query at inference. We also include results for prepending the entire context to the query (denoted as "GenerativeAdapter + Context") for comparison. The results show how combining our approach with context information at inference can enhance performance:
> > >
> > > | Context Length                | 512          | 1K           | 2K           | 4K           | 8K           | 16K          | 32K          |
> > > |-----------|------|------|------|------|------|------|------|
> > > | GenerativeAdapter                | 51.5 (0K)    | 49.3 (0K)    | 44.7 (0K)    | 40.9 (0K)    | 36.7 (0K)    | 32.7 (0K)    | 32.0 (0K)    |
> > > | GenerativeAdapter + Context      | 67.8 (0.5K)  | 67.8 (1K)    | 61.1 (2K)    | /            | /            | /            | /            |
> > > | GenerativeAdapter + RAG          | 61.9 (0.1K)  | 63.6 (0.1K)  | 60.8 (0.1K)  | 60.9 (0.1K)  | 60.1 (0.1K)  | 59.1 (0.1K)  | 56.5 (0.1K)  |
> > >
> > > **Additional Experimental Results (Section 4.2)**
> > >
> > > We expanded the comparisons in Section 4.2. Although we previously shared these results in our first reply, we are providing them again below for clarity:
> > >
> > > | Method                | Classification | Non-classification |
> > > |---------|------|-------|
> > > | Ultragist (256 tokens) | 41.1             | 7.5                  |
> > > | In-context prompting | 60.5           | 6.7                 |
> > > | Finetune            | 71.8           | 10.5                |
> > > | GenerativeAdapter  | 63.7           | 14.9                |
> > >
> > > These results show that Ultragist performs worse than in-context prompting on classification tasks and slightly better on non-classification tasks. GenerativeAdapter, on the other hand, achieves higher scores than both Ultragist and in-context prompting across a range of tasks, highlighting its effectiveness.
> > >
> > > Thank you again for your thoughtful feedback. We hope these additional experiments and clarifications address your concerns. We are happy to conduct further analyses if needed. If you find that the changes and clarifications made have satisfactorily resolved your queries and enhanced the quality of the work, we would greatly appreciate it if you could reflect this in your overall evaluation.

---

> > > ### Author Response · Authors · 2024-12-02
> > >
> > > Thank you so much for your time and valuable feedback. We would like to kindly remind you that Dec 2nd is the final day for reviewers to post messages to the authors. If you have any additional questions about the paper or concerns regarding gisting methods, please let us know at your earliest convenience.
> > > If you feel that the changes and clarifications made have sufficiently addressed your queries and improved the quality of the work, we would greatly appreciate it if you could consider this in your overall evaluation.

---

### Author Response · Authors · 2024-12-03
**Summary of the Discussions**

Dear AC,

Thank you so much for handling our paper! We would like to summarize the discussion at the end of the discussion phase.

To address the reviewers' concerns, we have made the following updates and clarifications:

1. **Baseline Methods (1mHi):**  As requested by the reviewer, we have included new results for Ultragist, a context compression method using gist tokens, in Sections 4.1 and 4.2.
2. **Perplexity as Validation Metric (mTD9):**  We added new results on document-based QA performance as part of the ablation study.
3. **Impact on Original Language Model Capabilities (mTD9):**  We provided new experimental results on MMLU to evaluate knowledge memorization across a wide range of domains, comparing models with and without the generated adapter.
4. **Data Inefficiency of the Continual Pretraining Baseline (1mHi, mTD9):**  We explained the low performance of the continual pretraining baseline in document-based QA, citing insights from prior research.
5. **Related Works (1mHi, PYAH):**  We acknowledged the relevance of hypernetworks and memory-based methods to our study. These connections were discussed in our response and will be elaborated in the paper.
6. **Methods and Implementation Details (apSm, PYAH):**  Reviewers noted minor issues in the description of methods and implementation. We will refine these sections for greater clarity in the final paper.
7. **Complementary Benefits on top of RAG (1mHi):** An LM merged with a generative adapter can be combined with prompting or other token compression approaches to provide complementary benefits. To demonstrate this, we conducted new experiments combining GenerativeAdapter with Retrieval-Augmented Generation (RAG), and the results show that this approach can further boost performance.

We summarized the strengths acknowledged by the reviewers and highlighted our contributions:

1. **Significance of the Problem (1mHi, apSm, mTD9, PYAH):**  Adapting language models to integrate new knowledge effectively and efficiently remains a critical challenge.
2. **Efficiency of Generating Adapters (1mHi, apSm, mTD9):**  The generative adapter mechanism introduced in our work is novel and reduces computational costs compared to fine-tuning. Also, by reusing previously generated weights, it supports streaming contexts and achieves efficient adaptation.
3. **Effective Knowledge Integration (PYAH):**  GenerativeAdapter is able to integrate factual knowledge into language model weights effectively on a single document. In contrast, conventional continual pre-training requires a large corpus and multiple repetitions of facts to achieve memorization.
4. **Miscellaneous Strengths:**  Reviewer 1mHi noted that the paper is clearly written and easy to follow.  Reviewer PYAH highlighted that our evaluation contexts are diverse and practical.

Thank you again for your time and efforts in reviewing our paper. We hope these updates address the reviewers' concerns and enhance the clarity of our contributions.

Best,

Authors

---

### Meta-Review · Area_Chair_xE8N · 2024-12-26

**Metareview:**

This paper introduces GenerativeAdapter, a novel technique for adapting pre-trained Large Language Models (LLMs) to new tasks without requiring fine-tuning or prompting.  Instead of modifying the core LLM parameters, GenerativeAdapter trains a lightweight adapter module using a self-supervised generative network. This adapter can process a continuous stream of input contexts, enabling the transformer model to adapt on-the-fly. This approach offers significant advantages in terms of efficiency, reducing both training time (compared to fine-tuning) and inference time (compared to prompting).

The effectiveness of GenerativeAdapter is demonstrated through experiments on various tasks, including factual knowledge acquisition, learning from demonstrations, and user personalization.  While reviewers have praised the significance and novelty of this work, they have also raised concerns about the chosen baseline methods and potential similarities to existing techniques like hypernetworks and memory-based methods.

To strengthen the paper, I recommend the following additions:

Comprehensive Baseline Comparisons: Include a broader range of baseline methods to thoroughly evaluate the performance of GenerativeAdapter.
In-Depth Analysis: Provide a detailed analysis comparing and contrasting GenerativeAdapter with hypernetworks and memory-based methods, highlighting its unique contributions and advantages.

**Additional Comments On Reviewer Discussion:**

After rebuttal, reviewers still have some concerns on baselines and relevance to other similar works.

---

### Decision · Program_Chairs · 2025-01-22

Accept (Poster)